# Circulating miRNAs as a Tool for Early Diagnosis of Endometrial Cancer—Implications for the Fertility-Sparing Process: Clinical, Biological, and Legal Aspects

**DOI:** 10.3390/ijms241411356

**Published:** 2023-07-12

**Authors:** Roberto Piergentili, Giuseppe Gullo, Giuseppe Basile, Caterina Gulia, Alessandro Porrello, Gaspare Cucinella, Enrico Marinelli, Simona Zaami

**Affiliations:** 1Istituto di Biologia e Patologia Molecolari del CNR (IBPM-CNR), 00185 Rome, Italy; roberto.piergentili@cnr.it; 2Department of Obstetrics and Gynecology, Villa Sofia Cervello Hospital, IVF UNIT, University of Palermo, 90146 Palermo, Italy; gaspare.cucinella@unipa.it; 3IRCCS Orthopedic Institute Galeazzi, 20161 Milan, Italy; basiletraumaforense@gmail.com; 4Dipartimento di Urologia, Ospedale della Misericordia, 58100 Grosseto, Italy; 85cate@live.it; 5Lineberger Comprehensive Cancer Center & RNA Discovery Center, University of North Carolina at Chapel Hill, 450 West Drive, Chapel Hill, NC 27599, USA; aporrelloresearch@yahoo.com; 6Department of Medico-Surgical Sciences and Biotechnologies, “Sapienza” University of Rome, 00161 Rome, Italy; enrico.marinelli@uniroma1.it; 7Department of Anatomical, Histological, Forensic and Orthopedic Sciences, “Sapienza” University of Rome, 00161 Rome, Italy; simona.zaami@uniroma1.it

**Keywords:** endometrial cancer (EC), assisted reproductive techniques (ART), miR-based, molecular evaluation, ethical and legal implications

## Abstract

This review article explores the possibility of developing an integrated approach to the management of the different needs of endometrial cancer (EC) patients seeking to become pregnant. Life preservation of the woman, health preservation of the baby, a precocious and—as much as possible—minimally invasive characterization of the health and fertility parameters of the patient, together with the concerns regarding the obstetric, neonatal, and adult health risks of the children conceived via assisted reproductive techniques (ART) are all essential aspects of the problem to be taken into consideration, yet the possibility to harmonize such needs through a concerted and integrated approach is still very challenging. This review aims to illustrate the main features of EC and how it affects the normal physiology of pre-menopausal women. We also focus on the prospect of a miR-based, molecular evaluation of patient health status, including both EC early diagnosis and staging and, similarly, the receptivity of the woman, discussing the possible evaluation of both aspects using a single specific panel of circulating miRs in the patient, thus allowing a relatively fast, non-invasive testing with a significantly reduced margin of error. Finally, the ethical and legal/regulatory aspects of such innovative techniques require not only a risk-benefit analysis; respect for patient autonomy and equitable health care access allocation are fundamental issues as well.

## 1. Introduction

Assisted reproductive technology (ART) refers to several techniques allowing women to achieve pregnancy in a non- (fully) natural way and include in vitro fertilization (IVF), intracytoplasmic sperm injection (ICSI), gamete intrafallopian transfer (GIFT), and preimplantation genetic diagnosis (PGD) [1]. ART is a viable option not only in couples with fertility impairment but also in women diagnosed with cancer set to undergo chemotherapy; such therapeutical approach, however, not only affects fertility by impairing the viability of gametes but could also alter their genome introducing deleterious mutations in the embryo. As such, ART is also a valuable option for EC patients to minimize those problems [2].

Endometrial cancer (EC) affects the lining of the uterus and is usually diagnosed in postmenopausal women, but about 5% of cases occur in women under 40, and approximately 20% of cases are diagnosed before menopause [3]. The most common type of endometrial cancer in premenopausal women is estrogen-sensitive adenocarcinoma (type I carcinoma), which has a better prognosis than more aggressive type II carcinoma. Several genes that play a central role in EC development and growth have been recognized in the last years, [4,5]; more recently, the epigenetics of EC has started to be unveiled. Micro RNAs (miRNAs or miRs) are short molecules of non-coding endogenous RNA that function as post-transcriptional regulators of gene expression [6]. In addition, in EC, miRs can play a role in the development and progression of cancer by modulating the expression of oncogenes and tumor suppressor genes. For example, some miRNAs have been reported to regulate the expression of PTEN, a tumor suppressor gene that is frequently mutated or deleted in EC (see below). In addition, miRs can also be associated with EC risk factors, such as insulin resistance and hyperinsulinemia, which can trigger cell proliferation and angiogenesis [7]. It is crucial to understand the molecular bases of EC and how it influences women’s fertility; of similar importance is the evaluation of who, how, and when can face oocyte preservation for ART procedures once EC is in remission.

Evaluation of female fertility is an important step for women who want to undergo ART or preserve their fertility after being diagnosed with EC. A pelvic examination should be performed to assess the uterus and adnexa for masses or other abnormalities. The gold standard method for diagnosing EC is hysteroscopy and endometrial biopsy, which have high sensitivity and specificity. Other imaging techniques, such as ultrasound, magnetic resonance imaging (MRI), or computed tomography (CT), can be used to evaluate the disease extent and plan the treatment [8].

In this work, we will examine all the above aspects. We will discuss the etiopathogenesis of EC both from a clinical and a molecular point of view, with a special focus on epigenetic causes and the use of circulating miRs as diagnostic biomarkers. We will report the state-of-the-art in fertility evaluation by means of circulating miR, comparing the two sets of data (circulating miRs in EC and for fertility assessment) and highlight how these two sets have only minimum overlap, providing the possibility to establish both the health status of the patient and her possibility to undergo ART at the same time and with a minimally invasive test (a blood sample). Finally, we will provide an overview of the main ethical and legal aspects of procreation in EC (and, more widely, in oncologic) patients. We set out to elucidate in a succinct and yet comprehensive fashion the underlying factors, determinants, and dynamics at the root of EC, its clinical implications in terms of therapeutics and diagnostics, the prospects for fertility preservation in younger patients, and the prospect for a miR-based, molecular evaluation of patient health status, including both EC early diagnosis and staging and, similarly, the patient’s receptivity. Finally, the complex ethical and legal/regulatory aspects have been weighed. For each of the above-listed subjects, 125 sources have been drawn upon, by using the search strings “endometrial cancer”, “fertility preservation”, “RNA-based molecular classification/evaluation”, “miRNA cancer diagnosis” through scientific databases PubMed/MedLine, PubMed Central, Scopus, ResearchGate, Web of Science. Only articles accounting for EC, infertility/fertility preservation, RNA-based diagnostic and therapeutic approaches, and ethics peculiarities of such innovative techniques have been considered for the fundamental purpose of this review article.

## 2. Etiology and Pathogenesis of EC in Fertile Women

### 2.1. Clinical and Endocrinological Characteristics of EC

EC has long been known as one of the most widespread gynecologic cancers worldwide; it is also the most common cancer affecting the female genital tract in developed countries. This malignancy is localized to the uterus in most patients (reportedly, as many as 67%) [9].

While uterine corpus cancer is currently the most common gynecologic malignancy, endometrial carcinomas constitute the majority of such diagnoses; sarcoma accounts for less than 10% of uterine corpus cancers. As many as 83% of uterine corpus cancers are endometrioid carcinomas. In addition, 4% to 6% of endometrial carcinomas consist of serous and papillary serous carcinomas, while clear cell carcinomas account for 1% to 2%. For a thorough analysis and better management and prevention of such conditions, it is worth drawing a distinction among type 1 endometrioid, type 2 serous endometrial carcinomas, and other highly aggressive non-endometrioid carcinoma histotypes [10,11].

Abnormal uterine bleeding or postmenopausal bleeding constitutes typical EC presentations. A diagnostic evaluation should be made available to any patient having EC risk factors, which should include the assessment of clinical history, imaging, and endometrial sampling. Standard EC therapeutic pathways may entail hysterectomy, bilateral salpingo-oophorectomy, and surgical staging. Hysterectomy is instrumental in accurately assessing EC prognostic factors, such as stage, grade, myometrial invasion, lymphovascular space invasion, and lymph node status [12,13,14].

ECs have been found to begin as preinvasive intraepithelial lesions transitioning into full-blown invasive cancers affecting endometrial stroma. A progressive penetration into the myometrium occurs through the lymphatic capillaries, thus spreading cancerous cells into regional lymph nodes. Then, the metastasizing process unfolds via vascular channels. The uterine cervix and stroma are likely affected by tumor progression through lymphatic channels, even if surface spread has been observed to take place from ECs manifesting in the lower uterine segment (LUS). LUS involvement in endometrial carcinoma has often been reported to result in lower survival rates and higher recurrence rates [15].

High levels of free estrogens leading to endometrial hyperplasia have been linked to estrogen-secreting, ovarian tumors, and polycystic ovaries (PCO); both conditions can adversely affect regular ovulation and menstruation. While anovulation obviously results in infertility, nulliparity is linked to a higher EC risk, even after adjusting for infertility [16]. 

Potential precursors of type I EC (which has been linked to a tumoral environment with excess estrogen) have been found to be atypical endometrial hyperplasia or endometrial intraepithelial neoplasia (EIN). Such dynamics often manifest at an early stage, with rather favorable outcomes. Serous, clear cell, mixed cell, and undifferentiated histologies are all elements reportedly associated with type II ECs, which are estrogen-independent and manifest at an already advanced stage with unfavorable prognosis [17]. The validation of genes or biomolecular factors is instrumental for an accurate prognosis assessment [18]. 

Such clinical dynamics entail medical as well as ethical and social concerns: fertility-sparing treatment (FST) can in fact forgo more radical care procedures by prioritizing the patient’s reproductive capabilities. ARTs are often required to that end, which means that EC treatment in patients of reproductive age is uniquely challenging, due to the need to strike a balance between “competing interests” of cancer care and the determination of patients to maintain their reproductive potential. Early menarche and late menopause, with higher levels of lifetime exposure to endogenous estrogens, have been found to lead to higher EC risks.

### 2.2. Challenges Arising from Fertility-Sparing Approaches in EC Patients

EC type and fundamental traits ought to be thoroughly assessed in order to choose the therapeutic pathway, which best suits each patient, particularly when weighing a conservative management opportunity.

Fertility-sparing procedures need to be weighed and counseled when making treatment decisions. That is even truer in light of the potentially harmful psychological dynamics that may be triggered by the loss of fertility following aggressive therapeutic approaches [19,20].

EC risk of extrauterine spread is an essential aspect to evaluate when assessing patient eligibility for fertility-sparing procedures. For patients who have an interest in preserving their fertility and plan to conceive as soon as possible after remission, fertility-sparing should be always considered, in the absence of contraindications and when there is favorable histopathological cancer makeup [21].

The fertility-sparing decision-making process needs to be weighed against various EC risk factors, such as obesity and polycystic ovary syndrome; such factors are linked to infertility as well; hence, any ART consideration may well be influenced by them. EC stage Ia grade 1 (G1) and EEC are the malignancies for which fertility-sparing treatment is most often chosen. EC type II, on the other hand, often makes patients ineligible for conservative treatments, due to its high level of invasiveness and poor differentiation. Young women with G1, no myometrium and/or adnexal invasion, and without lymphvascular space involvement, are therefore deemed the best candidates for fertility preservation approaches [22,23]).

When outlining any such pathway, it is worth taking into account updated guidelines by scientific societies such as ESGO-ESHRE-ESGE (the European Society of Gynaecological Oncology, the European Society of Human Reproduction and Embryology, and the European Society for Gynaecological Endoscopy, respectively), which have issued specific, evidence-based guidance for fertility-sparing treatment of EC patients, by focusing on the fundamental traits of fertility-sparing treatments. Particularly relevant is the recommendation that EC patients undergoing fertility-sparing procedures are counseled and cared for by a multidisciplinary team relying on oncologists and fertility specialists [24].

Progestin therapies (e.g., medroxyprogesterone acetate, MPA), megestrol acetate (MA), and progesterone-releasing intrauterine device (IUD) are the most widespread and validated EC hormonal treatment (HT) options and constitute the bedrock of the conservative fertility-sparing toolbox. MPA at 250–600 mg daily and MA at 160–480 mg daily are the most widely used regimens and rely on similar potency levels [25]. A 2016 meta-analysis by Qin et al., accounting for 25 sources comprising 445 women with early-stage EC treated with an oral progestin, has found an 82.4% regression rate, a 25% relapse rate, and a 28.8% pregnancy rate. Such findings point to the high degree of safety of oral progestins for early-stage EC patients who wish to have their fertility preserved [26].

Recent data have pointed to the novel levonorgestrel intrauterine device (LNG-IUD) as a solid fertility preservation option as well [27]. This device elicits a local hormonal surge in higher amounts and has efficacy rates similar to oral formulation, although conclusive comparative studies are not yet available [28].

## 3. Genetic and Epigenetic Factors Causing EC Pathogenesis

### 3.1. The Genetics of EC

As described before, pathological and demographic parameters allow the identification of two different types of EC: type I EC, also called EEC (endometrial endometrioid carcinomas), usually well differentiated, with average recurrence rates of 20%; and type II EC, which include mainly CCEC (clear cell endometrial carcinoma) and SEC (serous endometrial carcinoma). It is essential to differentiate type I from type II carcinomas and other highly aggressive non-endometrioid carcinoma histotypes to understand, manage and possibly prevent these diseases [18]. The molecular characterization of EC can facilitate the identification of various tumor subtypes. Approximately 80% of type I EC show expression downregulation or mutations of the phosphatase and tensin (*PTEN*) gene, which acts on the PI3K/Akt/mTOR signaling pathway, essential for the regulation of the cell cycle and is involved in cell survival, proliferation, and growth [29,30]. Several ongoing studies are currently aimed at inhibiting this pathway in advanced or recurrent EC [31] since the incidence of EC has been increasing, while the survival of EC patients did not significantly improve over the past 30 years [32]. Additional impaired cell cycle controls in EC include the RAS-RAF-MEK-ERK and canonical WNT-β-catenin pathways, both involved in the regulation of cell proliferation, cell survival, and differentiation [33]. Back in 2013, the TGCA Research Network reported the data collected from the genomic, transcriptomic, and proteomic analysis of 373 endometrial carcinomas, of which 307 were EECs, 53 were SECs, and 13 were mixed cases [34]. Those data allowed a classification into four classes of EC with distinct clinical, pathologic, and molecular features: (i) POLE (polymerase epsilon)/ultramutated (7% of cases), (ii) microsatellite instability (MSI)/hypermutated (28%), (iii) copy-number low/endometrioid (39%), and (iv) copy-number high/serous-like (26%). *POLE* gene encodes the catalytic subunit of DNA polymerase epsilon whose function is implicated in nuclear DNA replication and repair. *POLE* mutations usually occur in the soma and because of their function, have an extremely high mutation rate but also by excellent prognosis with no recurrence regardless of the FIGO grade. The MSI group also shows a high mutation rate and is mostly caused by *MLH1* promoter methylation. *MLH1* is a DNA mismatch repair protein, and its gene mutations are frequently associated with microsatellite instability (both facts explaining the high mutation rate in EC) as also observed in hereditary nonpolyposis colorectal cancer (HNPCC, or Lynch syndrome). *MLH1* mutations cause an EC risk of 25–60% [35] and result in an intermediate prognosis. The copy-number low/endometrioid group of EC shows a lower mutation rate compared to the previous two, no mutations in *POLE*, absence of MSI, and a low incidence of somatic copy-number variations. This genetic scenario is frequently found in low-grade EEC and the prognosis is intermediate. Finally, the fourth class is characterized by a low mutational rate but a high frequency of copy number variations and a 90% rate of TP53 mutations. This group mainly includes SEC and its patients, who unfortunately have a poor prognosis [36]. Recently (December 2020), the European Society of Gynaecological Oncology (ESGO), the European Society for Radiotherapy and Oncology (ESTRO), and the European Society of Pathology (ESP) published updated EC guidelines on the basis of the TGCA consortium findings [37]. To date, more than 300 genes mapping throughout the genome (with the only exception of the Y chromosome) and two mitochondrial genes have been studied for their role in EC [33]. According to the study of Bianco and co-workers, the most frequently mutated genes in endometrioid carcinomas are *PTEN* (>77%), *PIK3CA* (53%), *PIK3R1* (37%), *CTNNB1* (36%), *ARID1A* (35%), *K-RAS* (24%), *CTCF* (20%), *RPL22* (12%), *TP53* (11%), *FGFR2* (11%), and *ARID5B* (11%). The most frequently mutated genes in serous carcinomas are *TP53* (80–90%), *PIK3CA* (41.9%), *PPP2R1A* (36.6%), *FBXW7* (30.2%), *CHD4* (16.3%), *CSMD3* (11.6%), and *COLA11* (11.6%), along with loss of heterozygosity (LOH) on many chromosomes. MSI is frequent in type I carcinomas (25–40%) but relatively rare in type II carcinomas (<5%). Interestingly, *PTEN* mutations are characteristic of type I EC, while *TP53* mutations are characteristic of type II EC, affecting just 11% of EEC. Instead, *PIK3CA* mutations are a common characteristic of both types, being mutated in 40–50% of all EC. This finding suggests that, apart from some shared pathways, types I and II have a very distinct molecular pathogenesis. Similarly, the above-mentioned four subtypes of EC are molecularly different as well, with candidate driver or pathogenic genes accounting for 190 genes in the POLE subgroup, 21 in the MSI subgroup, 16 in the low copy number subgroup, and 8 in the high copy number subgroup [33]. In these subtypes, *PTEN* is the most frequently mutated gene in all except in the copy number high group, where TP53 mutations dominate. Additional genetic causes of EC formation and growth include single nucleotide variants in specific genes [38], shortened telomeres [39], and epigenetic factors, such as DNA hypermethylation of target gene promoters (as for *MLH1*) [40,41] or deregulation of both long and short non-coding (nc) RNAs [42].

In recent years, increasing interest has been devoted to these last molecules and their cross interaction, because of their great potential as EC biomarkers. Focused ncRNA panels are under investigation for creating fast and accurate diagnostic tools; with the aim of being as fast and accurate as possible, but at the same time without using invasive approaches, circulating short ncRNA attracted the attention of researchers.

### 3.2. Role of Circulating miRNA in EC: miR, ceRNET, and Cancer Biology

Micro-RNAs (miRNAs or miRs) are a class of single-stranded, non-coding RNAs (ncRNA) characterized by their short length: 20–25 nucleotides on average. They are part of a group of molecules collectively called ‘short non-coding RNAs’ (sncRNA). Their action is modulating gene expression by targeting messenger RNAs (mRNA) through sequence homology. Notably, the match between miR and its target is not always perfect, thus allowing the former to interact with multiple mRNAs; this makes the target discovery challenging and the function identification puzzling [43]. The binding miR/mRNA occurs in most cases at the 3′UTR end of the mRNA, and the final effect is typically the inhibition of the mRNA function, either by impairing its translation or by promoting its degradation [44]. Consequently, many miRs can be functionally considered gene expression silencers. It is estimated that more than 2500 miRs are encoded in the human genome, regulating over 60% of human genes [45]. Several metabolic pathways are controlled by miR action, including those involved in cell cycle control. In fact, the central role of miRs is now widely recognized in many cancers [46], including EC [47,48,49], where miR controls pivotal steps of tumor biology, such as proliferation, apoptosis, and invasion. Interestingly, miRs can have either an oncogenic or a tumor-suppressive action during cell cycle control, depending on several factors including (i) the role of the gene encoding the target mRNA; (ii) their own regulation, since miRs as well can be up- or down-regulated in different patients; (iii) their expression in different tissues. EC is not an exception: recent systematic reviews highlighted the role of the dysregulation of more than 100 miRs in the etiopathogenesis of EC [50,51]. Therefore, miRs are considered both valuable markers and very promising targets in cancer therapy [52]. However, the action mechanism of miR on mRNA is neither simple nor straightforward. First, as said, multiple miRs can target the same mRNA, and a single miR can target multiple mRNAs. Second, another heterogeneous class of non-coding RNA called long non-coding RNAs (lncRNA), is involved as well in this control. These molecules are longer ncRNA (200 nt or more), heterogeneous in several aspects (length, shape, cytological localization, and genome localization), and able to bind several miRs at the same time, a phenomenon called ‘miR sponging’ [53]. In the last years, hundreds of papers have been published describing single components of these control pathways, the sum of one lncRNA, one sncRNA, and one target mRNA interacting with each other being called an “axis”. In these axes, mRNA, and lncRNA compete for miR binding, and multiple, interconnected axes create what is now commonly known as a ceRNET (competing endogenous RNA network). In this network, nodes are ceRNAs (competing endogenous RNAs, i.e., lncRNA and mRNA), while miRs represent their connections [54,55]. This complex epigenetic gene expression control allows very fine tuning of cell cycle regulation. Despite this organization should provide a certain redundancy able to confer robustness to mutations, it also allows a single alteration to hit multiple metabolic pathways at the same time, hence, the importance to identify and characterize ceRNET. These networks have been partially reported in neurodegenerative diseases [56,57] and in a few cancers such as lung adenocarcinoma [58], intrahepatic cholangiocellular carcinoma [59], hepatocellular carcinoma [60], glioma [61], thyroid cancer [62], and breast cancer [63,64]. However, for most cancer types, only single axes have been published so far. Recently, a complex ceRNET involving the lncRNA encoded inside the X-linked chromosome inactivation center and involved in multiple organ carcinogenesis has been described [65].

The literature reports several ceRNA axes in EC [66] and a long list of simpler interactions has been described as well, with only two components for each putative axis. Recently, ceRNET started to be described in EC as well. In 2019, Zhao and collaborators analyzed original data of EC RNA transcripts from The Cancer Genome Atlas (TCGA) database in search of prognostic biomarkers in endometrial carcinoma [67]. This study allowed the identification of 62 lncRNA, 26 miR, and 70 mRNA deregulated in EC. Amongst them, 10 lncRNA, 19 mRNA, and 4 miRs were closely associated with the survival of EC patients (*p* < 0.05). In 2022, Cai and coworkers analyzed both the TCGA and the Clinical Proteomic Tumor Analysis Consortium (CPTAC) databases to reconstruct a lncRNA-mediated ceRNA network for uterine corpus endometrial carcinoma [68]. This work allowed identifying tens of these axes, with six of them carrying a high prognostic value. In the same year, Song and collaborators also took advantage of TGCA database to reconstruct a ceRNET including 5 deregulated lncRNA, 7 deregulated miR, and 90 deregulated mRNA [69]. In Figure 1, we report two simplified examples of ceRNET in EC.

### 3.3. Using Circulating miRs for EC Diagnosis

According to recent advances in histopathological and molecular characterization, EC can be classified into at least four different classes [78,79]. In addition, adequate biomarkers may also be used as valuable prognostic tools [80], and miR might be promising tools during patients’ evaluation [81]. The accurate characterization of each patient is fundamental for their management, especially when it is desirable to preserve fertility [37,82]. In this perspective, the use of adequate biomarkers and their early detection, possibly with a fast and minimally invasive approach, is very important. It is possible to characterize EC according to the expression of different miR—and, consequently, their ceRNET-related characteristics [47,48,49]. Many studies have detected extracellular/circulating miR in biological fluids, such as plasma and serum, cerebrospinal fluid, saliva, breast milk, urine, tears, colostrum, peritoneal fluid, bronchial lavage, seminal fluid, and ovarian follicular fluid (see [6] and references therein). In these fluids, miRs can be either detected inside extracellular vesicles (exosomes, microvesicles, and apoptotic bodies) or floating in association with proteins such as Ago2. Interestingly, miR stability in body fluids is far higher than inside cells [83,84]. The reason why miRs—which are generated inside cells—are present in body fluids is still unknown. Two main hypotheses are under investigation: they are merely a byproduct of extensive cell death (mainly apoptosis and necrosis) connected to tumor biology (a passive mechanism), or they fulfill long-distance, cell-cell communication tasks (an active mechanism).

The identification of deregulated miRs in EC patients is ongoing, and the discovery of new ceRNA axes is relatively frequent. A meta-analysis published in 2019 by Delangle and co-workers [85] identified the deregulation of the expression levels of 261 miRs in EC, divided into 133 onco-miRs, 110 miRs onco-suppressors, and 18 miRs with discordant functions; of them, 139 showed deregulation in endometrial tissue compared to benign and/or hyperplastic tissues (Figure 2). However, the possibility to use circulating miRs to achieve this characterization is still under evaluation and has been only poorly explored in EC. In 2022, Bloomfield and collaborators performed a systematic review in search of circulating miRs in the serum and plasma of EC patients [86]. Their analysis allowed identifying 33 significantly deregulated miRs (Figure 2), 27 up- and 4 down-regulated, while the remaining 2 (miR-21 and miR-204) showed contradictory expression values, depending on the study considered (Table 1). These works show that adequate combinations of miR expression values may be used as prognostic markers, able to help in defining EC histological type and grade, tumor size, FIGO stage, lymph node involvement, and survival rate, thus demonstrating the miR role in decision- making for patients’ management [85]. Interestingly, a simple one-to-one comparison of the two sets of data reported above indicates that almost half (16/33) of the described circulating miRs are also deregulated in EC specimens. As shown in Table 1, while in most cases up- or down-regulation matches, in both reports, in a few cases (miR-9, miR-99a, miR-100, and miR-199b), they are discordant. Considering the two hypotheses described above, concordant regulation might indeed be a passive mechanism due to cell death and the consequent release of cellular debris in the bloodstream. However, discordant results might allow for the consideration of an active mechanism that, if further supported by the data, might shed light on new and interesting features of miR biology in EC.

Eismann and coworkers in 2017 showed that the circulating miRs excreted in vitro by EC cell lines are also dependent on exogenous environmental stress, such as hypoxia and acidosis [111]. They showed that the hypoxia caused the downregulation of miR-15a, miR-20a, miR-20b, and miR-128-1 in Ishikawa cells (type I EC) and the upregulation of miR-21 in EFE-184 cells (type I EC), while acidosis caused the upregulation of the oncogenic miR-125b in AN3-CA cell (type II EC), while in Ishikawa cells (type I EC), miRs with tumor suppressive function were found altered in divergent directions, either up- (let-7a) or down- (miR-22) regulated. These data show that at least in type I EC cells, hypoxia promotes the downregulation of secreted miRs with tumor suppressive and anti-angiogenetic function and the contemporary upregulation of secreted miRNAs with tumor and angiogenesis-promoting function. Instead, acidosis caused the upregulation of tumor-promoting miRs in type II EC. Collectively, these data suggest that the miR profile not only might be used to identify EC in patients but also to evaluate the status of the tumor microenvironment and its changes over time.

## 4. Profiling the miR Transcriptome for the Evaluation of Endometrial Receptivity

To date, only a minor proportion of studies analyzed the expression levels of miRs in healthy women to evaluate their potential receptivity.

In 2017, Altmae and collaborators performed a meta-analysis using as ‘bait’ a meta-signature of endometrial receptivity involving 57 mRNA genes as putative receptivity markers [112]. Searching for putative regulators and using the robust rank aggregation (RRA) method [113], they identified 19 miRs with 11 corresponding up-regulated meta-signature genes. Of them, three (miR-30c-1, miR-130b, and miR-449c-5p) are also linked to human EC [98,99,114]. Drissennek and collaborators performed a retrospective analysis of the miRNome of endometrium samples collected during the implantation and associated with the receptivity status and the pregnancy outcome (implantation failure, early embryo miscarriage, and live birth at term) in patients with a positive or negative b-hCG [115]. They identified 11 miRs associated with the endometrial receptivity status (9 downregulated); among them, miR-455-3p was also a putative tumor suppressor. They also showed that the overexpression of miR-152-3p and miR-155-5p in receptive endometrium is associated with implantation failure, with both miRs being involved also in the etiopathogenesis of several types of cancer. Unfortunately, additional validation tests did not allow producing more reliable results on other miRs identified during the study, including circulating miR. Additional research identified other miRs involved in these processes, such as miR-135b [116], let-7-a [117], miR-21 [118], miR-22 [119], miR-23b and miR-145 [120], miR-30b/d [121,122], miR-148a-3p [123], miR-181a/b [124,125], miR-194-3p [126], miR-200a [127,128], miR-429, miR-4668 and miR-5088 [129], and miR-494 [121]. Only a few additional studies investigated circulating miRs related to women’s fertility, but despite their limited number, they allowed identifying also miR-25, miR-27a, miR-31, miR-93, miR-106b, miR-146a, miR-152, and miR-155 (reviewed in [130]). 

It is noteworthy to underline that, among all these fertility-related miR, only miR-21, miR-27a, and miR-135b are present in Table 1, i.e., they are circulating miRs related to EC. If further validated, these data would allow for the characterization of the presence of EC and the fertility status of the patients using distinct miR panels, without ambiguity and relying on the collection of blood samples only.

## 5. RNA-Based Diagnostics and Therapeutics: Are Innovations Set to Outpace Bioethics Precepts?

RNA-based measurements are potentially applicable through a wide array of medical areas, including diagnosis, prognosis, and therapy selection. Currently, among the most promising clinical applications, it is worth mentioning not only cancer research [131] but also infectious diseases, transplant medicine, and fetal monitoring [132]. RNA sequencing (RNA-seq) has enabled us to detect a remarkably wide array of RNA species, such as messenger RNA (mRNA), non-coding RNA, pathogen RNA, chimeric gene fusions, transcript isoforms, and splice variants. RNA-seq has also led to the possibility of quantifying known, pre-defined RNA species and rare RNA transcript variants. Not only differential expression and detection of novel transcripts are possible, but also the detection of mutations and germline variation can be achieved through RNA-seq, possibly involving hundreds to thousands of expressed genetic variants, thus enhancing our ability to evaluate the allele-specific expression of these variants. Since the mechanisms governing RNA for diagnostics and therapeutics were first discovered and explored in the late 1990s, therapeutics based on RNA interference (RNAi, a mechanism for gene silencing underpinned by short interfering RNAs, siRNA, which was discovered in 1997 by Craig Mello) [133,134] have been developing remarkably fast, and our understanding of such highly complex processes and interactions has deepened considerably [135]. RNA-based clinical trials have already begun. Such therapies relying on “gene-silencing” are even more controversial than diagnostic applications since they may be viewed as akin to gene-editing/genetic engineering. It has been literally decades since such techniques have been developing and eliciting intense debates among scientists, bioethicists, policy- and law-makers, centered around how to best harness the potential of such breakthroughs for the benefit, potentially, of billions of human beings [136]. The hope and dream that diseases could one day be eradicated through the deliberate and targeted manipulation or editing of genes were the driving force behind the Human Genome Project [137,138], through which the complete human DNA sequence was first outlined and mapped in 2003. Such a fundamental principle, at the core of which lies disease treatment via genetic modification, dated back to the 1960s, i.e., when it was first observed that viral DNA had the ability to trigger cellular modulation during an infection. Early efforts aimed at gene modification were dated back to the 1970s [139,140]; recombinant DNAs (rDNAs), i.e., a combination of more than one DNA sequence from one or more species, were used for that purpose. Primary transfection methods were viral infection and calcium phosphate. Such innovations gradually bore fruit in the form of cell line development, genetically modified animals, and the creation of human proteins such as insulin in bacteria [141]. By the late 1970s, mRNA in vitro had been transfected by liposomes; this gave rise to rabbit globin expressed in mouse lymphocytes [142]. mRNA sequences in the cell cytoplasm, in order to inhibit protein translation or to induce exon skipping were targeted by newly developed antisense oligonucleotides (ASOs) [143]. Not surprisingly, it was back then that such fast-moving progress ignited a broad-ranging and concerted effort encompassing genetic engineering in terms of its ethical, social, political, and even economic implications. As a result, regulations limiting the different “tiers” of gene editing were drafted and enacted. Later, major developments are constituted by zinc finger nuclease (ZFN) to cleave a target DNA, and two decades later, TALEN and CRISPR/Cas9. Hence, it stands to reason that before such novel and potentially revolutionary therapeutic approaches can become mainstream from the standpoint of clinical applications, it is of utmost importance to discuss the legal and ethical issues arising from their use [144]. An analysis of the ethically relevant features of RNAi therapies is essential for the purpose of producing a comprehensive risk-benefit analysis. Ethically relevant traits such as siRNA delivery and the specificity of silencing effects cannot be swept aside. Furthermore, the future development of RNAi-based therapeutic options ought to consider and respect patient autonomy by accounting for the risks of generating infection-competent viruses or possibly introducing genetic changes in germ-line cells. Just as importantly, issues relative to justice in care delivery, such as equal access as opposed to private acquisition, and the right to participate in clinical trials should also be prioritized. The sheer scale of progress made in ncRNA research applied to cancer, and our ever-greater understanding of tumor biology, which will lay the groundwork for the development of new ‘smart’ drugs tailored to a patient-oriented approach, is poised to enable us to minimize adverse events and improve the patient prospects for recovery [145]. However, an important risk is that innovative biomedical techniques may outpace our ethical, legal, and regulatory frameworks, leading to “grey areas” similar to those found in genome-editing research and artificial intelligence. In addition, assisted reproductive technologies and fertility preservation (oncofertility) also pose complex and challenging ethical and legal issues. In fact, even though oncofertility procedures are not life-saving, they can undoubtedly be viewed as “life-enhancing” or “life-giving”. This characterization is fully supported by the wide-ranging conception of health, in adherence to the 1946 Constitution of the World Health Organization, which covers a broader notion of well-being that reaches way farther than the absence of disease [146]. In that regard, there is no discounting the fact that cancer-related or iatrogenic infertility following cancer treatment can indeed engender emotional and psychological implications that may lead to psychiatric conditions. Such a risk is hardly surprising, given the fact that a cancer diagnosis itself frequently poses major psychological strain [147], and perceiving the risk of infertility often adds to the mental and psychological distress, leading to low self-esteem, anxiety, depression, and a noxious sense of personal worthlessness, which have the potential to cause major deterioration and ignite psychiatric diseases. Hence, in light of the well-documented connection between infertility, health, and mental issues, the fundamental importance of counseling cannot be overstated [148]. Guaranteeing access to such care without discrimination and inequality is, therefore, a medical, moral, ethical, and legal imperative. However, despite their undisputed value, oncofertility procedures are ethically and ethically controversial much for the same reasons that ARTs are. Legislative frameworks regulating such techniques reflect that delicate balance. ARTs are governed at the national level with varying degrees of restrictions in Europe and globally. That is hardly surprising, considering the various social and moral principles that each society decides to prioritize through specific national norms. Beginning-of-life issues are certainly among the most complex and variously regulated matters overall. Although an in-depth comparative analysis is beyond the scope of this article, it is still worth mentioning that in the European Union, the European Court of Human Rights (ECHR) has over the years espoused a broad margin of appreciation [149] for member states when governing matters with fundamental social, moral, and ethical valence [150,151]. Fertility preservation for cancer patients serves a fundamental purpose that cannot be discounted: it is meant to discharge the moral duty to uphold the reproductive autonomy of individuals, an inalienable human right that any free society should strive to enforce [152]. It is certainly safe to assume that mentally capable adults should be enabled to exercise such rights, provided that no unreasonable risk arises for the children thus conceived or to others. As far as medicolegal implications are concerned, failing to provide information and counseling on fertility preservation opportunities might even constitute grounds for a negligence-based malpractice lawsuit for loss of chance, if the patient’s reproductive capabilities are provably impaired as a result, even though currently available case law on the matter is not yet entirely conclusive [153,154,155]. Against such a backdrop, it is therefore worth remarking on the essential role of reproductive counseling within the framework of a multidisciplinary management approach to cancer care. Patient consent must possess specific features: it must be fully informed, free, unequivocal, specific, and revocable at any time. It is the onus of the healthcare professional to document the accuracy of the whole process. Such key aspects have been highlighted by evidence-based recommendations and guidelines by leading international scientific organizations, such as the American Society for Reproductive Medicine [156], the American Society of Clinical Oncology [157,158], and the already mentioned ESHRE [159].

Fertility preservation should only be offered to patients with EC stage Ia grade 1 (G1), who present without myometrial invasion or where cancer has invaded less than 50% of the myometrium, with no evidence of pathological lymph nodes or synchronous/metachronous ovarian tumor [16]. The most common fertility-sparing treatment for EC is hormonal therapy with progestins, which can induce regression or stabilization of the tumor. However, this treatment has several limitations, such as a low response rate, high recurrence rate, lack of standardization, and potential adverse effects on the fetus. Therefore, patients who opt for fertility preservation should be carefully selected and counseled about the risks and benefits of this approach. They should also be closely monitored during and after the treatment and advised to undergo definitive surgery after completing their childbearing. ART can offer an alternative or complementary option for women who want to conceive after being treated for EC. However, ART also poses some ethical dilemmas, such as the safety and efficacy of the procedures, the potential harm to the mother and the child, the disposal or donation of surplus embryos [160], and the access and affordability of the services [161,162]. Moreover, some ART techniques involve genetic testing or manipulation of embryos, which raises further ethical questions about respect for human dignity, autonomy, and diversity [163,164]. Undoubtedly, ART can provide hope and opportunity for women who want to have children after being diagnosed with EC. However, ART also involves medical, legal, and ethical challenges that require careful consideration and multidisciplinary collaboration. Therefore, patients who are interested in ART should be informed and supported by a team of experts who can help them to make informed decisions that are consistent with their values and preferences [165].

## 6. Discussion

Although gynecological cancers are most frequently diagnosed in post-menopausal patients, substantial numbers of younger women of reproductive age are also affected. The importance of thorough counseling of these patients cannot be overstated, including a comprehensive discussion of long-term consequences on fertility of the various treatment options and aiming at fertility preservation as much as possible, if the patient prioritizes this aspect. EC diagnosis at reproductive age points to the risk of a hereditary condition; hence, counseling must address the genetic cancer risk evaluation in order to investigate any known genetic predisposition. Such broad-ranging principles notwithstanding, fertility-sparing therapeutic pathways need to be individually tailored, considering the high level of heterogeneity of gynecological cancers.

The chief surgical option for EC is hysterectomy, whereas fertility-sparing management in patients with EC or complex atypical hyperplasia is not deemed the standard approach. Eligibility requirements are age under 40, complex atypical hyperplasia, or grade 1 EC limited to the endometrium. Therefore, an accurate pretreatment staging is non-negotiable [166].

Gallos et al., in a meta-analysis involving 408 women with early-stage EC who underwent fertility-sparing treatments, found a 28% live birth rate, with patients undergoing ART achieving a 39.4% live birth rate, as opposed to 14.9% in the spontaneous conception group [167].

The preservation of fertility and procreative capabilities are ever-more relevant elements in terms of life quality after EC. Patients with no male partner or unwilling to freeze and store embryos can still preserve their oocytes. Such techniques, particularly those relying on freezing by vitrification, lower the risk of incurring multiple pregnancies or ovarian hyperstimulation syndrome (OHSS). Compared to transfer from fresh embryos, no significant differences have been reported in terms of pregnancy rate per cycle and the clinical pregnancy rate per cycle [168].

The decision to store frozen oocytes, as opposed to embryos, can avoid or at least alleviate ethical and legal challenges often arising from cryopreserved embryos and their controversial status. Major improvements in overall outcomes of oocyte cryopreservation have been achieved through more effective cryopreservation techniques such as vitrification rather than the slow-freeze protocol, mostly thanks to the reduction of cellular damage arising from ice crystal formation [169]. Moreover, it is worth remarking that most ECs are estrogen-dependent, which questions the safety of pregnancy and the risk of cancer recurrence related to it.

The intersection of gonadotoxic therapy and reproduction raises ethical issues for both cancer and fertility specialists, including issues of experimental vs established therapies, the ability of minors to give consent, the welfare of expected children, and posthumous reproduction [154,170,171].

The newly found possibility to focus both on the health status of the patient, including the definition of EC staging and her fertility, with a fast and minimally invasive approach, as described above, further contributes to enabling such patients to achieve motherhood, through effective management of their treatments.

## 7. Conclusions

As infertility after cancer has become a recognized survivorship issue, oncologists should be prepared to discuss the negative impact of cancer therapy on reproductive potential with their female patients in the same way as any other risks of cancer treatment are discussed. Patients interested in fertility preservation should be promptly referred to a reproductive medicine expert to offer timely and appropriate counseling and improve the success of fertility preservation. Such aspects are to be viewed as requirements for the medicolegal tenability of any intervention. Reproductive endocrinologists should collaborate with oncologists and molecular biologists, updating them regarding available technologies and facilitating consultations with patients newly diagnosed with cancer. To further these alliances, education about fertility preservation, as well as ethical and legal aspects tailored to country-specific laws, should be incorporated into training programs for oncology and reproductive endocrinology. Just as importantly, the breakthrough constituted by RNA-based diagnostics and therapeutics needs to rely on as broad a consensus as possible, by all stakeholders involved (patients, healthcare professionals, law- and policy-makers, bioethicists, and patient rights organizations) in order to reconcile the amazing opportunities created by such techniques with the core values that must guide scientific research and medical practice at all times. 

## Figures and Tables

**Figure 1 ijms-24-11356-f001:**
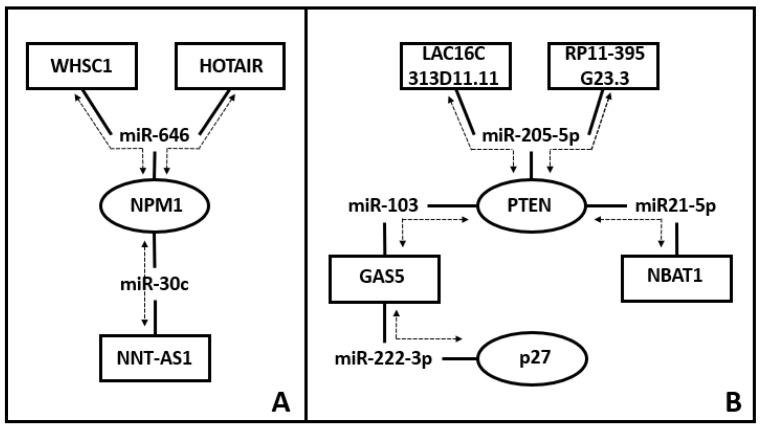
Examples of ceRNET in EC. The reported examples are only for exemplification purposes, and they are not exhaustive of EC-related ceRNET identified so far. Nodes are ceRNA (i.e., a lncRNA or an mRNA) and in the scheme, they are represented as either ovals (mRNA) or rectangles (lncRNA). Their connections (lines) identify miR interacting with both molecules (one oval, one rectangle). Individual axes (mRNA + miR + lncRNA) are identified by dotted lines with arrows and are the same retrieved by the available bibliography (see below). In panel (**A**), the two top axes share both a connection (miR-646) and a node (NPM1) and, at the same time, the node NPM1 is shared with the lower axis. In panel (**B**), a more complex situation is depicted: the two top axes share both a connection (miR-205-5p) and a node (PTEN), which is shared with two additional axes.; o One of these PTEN-linked axes (PTEN/miR103/GAS5) is in turn connected with an additional axis, sharing the node identified by GAS5. Bibliographic sources for these schemes are reported in the references list [70,71,72,73,74,75,76,77].

**Figure 2 ijms-24-11356-f002:**
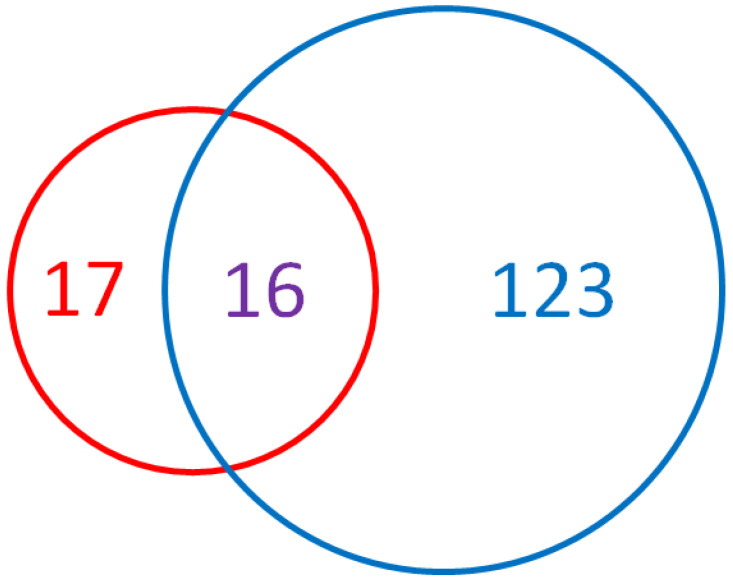
Comparison of miRs identified in the studies of Bloomfield and collaborators [86] and Delangle and collaborators [85]. The former study (red circle) identified 33 different miRs deregulated in EC; the second (blue circle) identified 139 miRs; 16 miRs are reported in both publications. These 16 miRs are further described in Table 1.

**Table 1 ijms-24-11356-t001:** Deregulated miRs in EC. The table reports the miRs that are in common between the studies of Bloomfield and collaborators [86] and Delangle et al. [85] as depicted in Figure 2. Arrows indicate the up- (upwards) or down- (downwards) regulation of miR; arrows pointing in opposite directions represent inconclusive results; extr stands for extracellular (i.e., serum/plasma-derived, data from [86]) miR; intr stands for intracellular (i.e., specimen-derived, data from [85] miR. Original reference(s) indicate the original manuscripts used by the two groups (Bloomfield and collaborators, Delangle and collaborators) for their literature analysis.

miR Name	extr.	intr.	Original Reference (s)
9	↓	↑	[87,88,89,90]
21	↑↓	↓	[91,92]
27a	↑	↑	[93,94,95]
30a-5p	↓	↓	[91]
99a	↑	↓	[96,97]
100	↑	↓	[88,96]
135b	↑	↑	[89,91]
141	↑	↑	[88,89,98,99]
142-3p	↑	↑↓	[100,101,102,103]
199b	↑	↓	[88,90,96]
200a	↑	↑	[88,89,90,98,104,105,106]
203	↑	↑	[88,89,98,99,107]
204	↑↓	↓	[99,108]
205	↑	↑	[87,88,89,91,98,99,104,105,109]
223	↑	↑	[87,93,98,110]
449	↑	↑	[89,99]

## Data Availability

All data are available upon request to the corresponding author.

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
