# Peer review of "Circulating miRNAs as a Tool for Early Diagnosis of Endometrial Cancer—Implications for the Fertility-Sparing Process: Clinical, Biological, and Legal Aspects"

_ijms, 2023, doi:10.3390/ijms241411356_

Round 1

Reviewer 1 Report

Below, I present my observations on the article titled 'Circulating miRNAs as a tool for early diagnosis of endometrial cancer – implications for the fertility-sparing process: clinical, biological and legal aspects.'

Major Observations

If the review focuses on miRNAs, there is no description of any molecular aspect that has been reported for the etiology and pathogenesis in fertile women. If the article deals with miRNAs and EC, it is crucial to describe the molecular mechanisms that have already been reported favoring the pathogenesis of this pathology before introducing miRNAs.

The writing strategy of placing 'discussion' as section 3 is not entirely clear to me. Although I consider that part well-written and providing valuable information, I believe this part should be a discussion within section 2.1. The importance of specific (molecular and laboratory) biomarkers and their relevance for early diagnosis should be discussed, as well as their relationship with viability and preservation of reproductive function.

The legal aspects related to fertility preservation in EC-diagnosed patients, including relevant laws and regulations, patient rights, ethical issues, and their application in different countries, should be discussed. Additionally, the legal challenges and barriers to accessing fertility preservation should be addressed, and possible solutions and recommendations to improve access and legal coverage should be speculated upon.

Figure 1 is not easily understandable, and too much information is included in the text that could be complemented within the figure itself.

Table 1 should be better represented and completed with the information indicated in the references, specifying the type of miRNA and where its main function has been found. I believe that the original references for each miRNA should be analyzed, giving credit to their respective original authors and examining the function of each overexpression or repression of those miRNAs.

Although the main problem with Table 1 is that the authors' last point has already been described by references (80) and (81). I really consider the contribution to be minimal.

Although the idea of the review article is promising, many aspects remain inconclusive. After reading it, there is a feeling that the idea is interesting but falls short in execution, and therefore, in this version, it is not possible to correlate the title with the main information.

Minor Observations.

Line 70, 72: abbreviate endometrial cancer as EC.

Line 148, 149: abbreviate assisted reproductive technologies as ART

Author Response

Dear Reviewer,

My coauthors and I are grateful for your great contribtion to the review process.

We have amended the manuscript according to your suggestions.

Specifically:

If the review focuses on miRNAs, there is no description of any molecular aspect that has been reported for the etiology and pathogenesis in fertile women. If the article deals with miRNAs and EC, it is crucial to describe the molecular mechanisms that have already been reported favoring the pathogenesis of this pathology before introducing miRNAs.

ANSWER: Since the article deals with miRNA and EC, we introduced the main molecular mechanisms of EC pathogenesis in a specific section just before introducing miR, as indicated (see pages 4-6). We thank the Reviewer for suggesting this text improvement.

The writing strategy of placing 'discussion' as section 3 is not entirely clear to me. Although I consider that part well-written and providing valuable information, I believe this part should be a discussion within section 2.1. The importance of specific (molecular and laboratory) biomarkers and their relevance for early diagnosis should be discussed, as well as their relationship with viability and preservation of reproductive function.

ANSWER: We have rearranged the structure to reflect your suggestion, with which we agree (see pages 4-6). we have also eliminated the Results section and replaced it with more straightforward subsections.

The legal aspects related to fertility preservation in EC-diagnosed patients, including relevant laws and regulations, patient rights, ethical issues, and their application in different countries, should be discussed. Additionally, the legal challenges and barriers to accessing fertility preservation should be addressed, and possible solutions and recommendations to improve access and legal coverage should be speculated upon.

ANSWER: More in-depth remarks have been laid out relative to the ethical, legal and social implications of oncofertility, in accordance with international legal perspectives and evidence-based guidelines on the subject (see pages 12-13, lines 557-603). Relevant sources have been added as well.

Figure 1 is not easily understandable, and too much information is included in the text that could be complemented within the figure itself.

ANSWER: The text in lines 287-301 was not changed. We simplified the figure legend by deleting unnecessary information. We also added dotted arrows to delimitate single axes, to better illustrate how different axes interconnect. We hope that now the figure is more readable and the legend easier to read and understand.

Table 1 should be better represented and completed with the information indicated in the references, specifying the type of miRNA and where its main function has been found. I believe that the original references for each miRNA should be analyzed, giving credit to their respective original authors and examining the function of each overexpression or repression of those miRNAs.

ANSWER: To better illustrate the data, a schematic figure (figure 2) has been added, explaining how miR reported in Table 1 were chosen. As for Table 1, it has been redrawn in the following way: (i) only overlapping miR were retained, since non-overlapping ones might confuse the reader; (ii) the original references from which the data were collected and analyzed in the systematic review and meta analysis were added in column 4. Where miR main function has been found is already present in columns 2 and 3, where ‘extr’ indicates extracellular sources (i.e., serum and plasma) and ‘intr’ indicates intracellular localization, i.e., specimen, biopsies and similar; this has been indicated in the legend, too. As already indicated, the arrows specify either up- or down-regulation of miR compared to healthy controls. We believe that an in-depth analysis of the molecular function of every single miR, as suggested, may defocus the aim of our review, in this contradicting the request of the Reviewer that notes that in the previous form, ‘it is not possible to correlate the title with the main information’. The aim of our work is to analyze the possibility of using circulating miR for the discrimination of EC type and for the contemporary evaluation of the fertility status of the women, independently of the role each miR plays inside (or outside) the cell. We hope that these additional explanations and the changes to the text will fulfill Reviewer’s requests.

Although the main problem with Table 1 is that the authors' last point has already been described by references (80) and (81). I really consider the contribution to be minimal.

ANSWER: We agree with the Reviewer that this information is unnecessary. We deleted the sentence accordingly. We thank the Reviewer for noticing this.

Although the idea of the review article is promising, many aspects remain inconclusive. After reading it, there is a feeling that the idea is interesting but falls short in execution, and therefore, in this version, it is not possible to correlate the title with the main information.

ANSWER: We hope that the improvements we introduced thanks to the help of the Reviewer enhanced the overall quality of the manuscript and make clearer the scope of the work to the reader.

Reviewer 2 Report

The manuscript by Piergentili et al. “Circulating miRNAs as a tool for early diagnosis of endometrial cancer – implications for the fertility-sparing process: clinical, biological and legal aspects” is a review focused mainly on the RNA-based diagnostics of endometrial cancer.

Manuscript is well-written, presented in an intelligible fashion and the language is clear and correct. Figure and table are clear and informative. Most references are up to date and appropriate.

Importantly, the authors clearly described the process of finding scientific papers upon which they based their review.

In my opinion, this manuscript is suitable for publication in International Journal of Molecular Sciences.

Author Response

Dear Reviewer,

We would like to thank you for taking the time to review the manuscript.

More importantly, we are grateful for and proud of your positive remarks.

We have attempted to improve the manuscript further, by rationalizing the structure and adding more substance to the legal and ethical part, among other things.

Warm regards,

Giuseppe Gullo, MD, PhD and coauthors

Round 2

Reviewer 1 Report

The authors have made significant changes addressing the comments, and the new version of the article is ready for publication. Perhaps just a little bit of English style correction.

Perhaps an English style editing service is needed.